# SARS-CoV-2 Is a Culprit for Some, but Not All Acute Ischemic Strokes: A Report from the Multinational COVID-19 Stroke Study Group

**DOI:** 10.3390/jcm10050931

**Published:** 2021-03-01

**Authors:** Shima Shahjouei, Michelle Anyaehie, Eric Koza, Georgios Tsivgoulis, Soheil Naderi, Ashkan Mowla, Venkatesh Avula, Alireza Vafaei Sadr, Durgesh Chaudhary, Ghasem Farahmand, Christoph Griessenauer, Mahmoud Reza Azarpazhooh, Debdipto Misra, Jiang Li, Vida Abedi, Ramin Zand

**Affiliations:** 1Neurology Department, Neuroscience Institute, Geisinger Health System, Danville, PA 17822, USA; sshahjouei@geisinger.edu (S.S.); MAnyaehie@som.geisinger.edu (M.A.); mowla@usc.edu (A.M.); vavula1@geisinger.edu (V.A.); dpchaudhary@geisinger.edu (D.C.); cgriessenauer@geisinger.edu (C.G.); 2Neuroscience Institute, Geisinger Commonwealth School of Medicine, Scranton, PA 18510, USA; EKoza@som.geisinger.edu; 3Second Department of Neurology, “Attikon” University Hospital, National and Kapodistrian University of Athens, School of Medicine, 12462 Athens, Greece; tsivgoulisgiorg@yahoo.gr; 4Department of Neurosurgery, Tehran University of Medical Sciences, Tehran 14155-6559, Iran; soheilnaaderi@gmail.com; 5Division of Stroke and Endovascular Neurosurgery, Department of Neurological Surgery, Keck School of Medicine, University of Southern California, Los Angeles, CA 90033, USA; 6Department de Physique Theorique and Center for Astroparticle Physics, University Geneva, 1211 Geneva, Switzerland; vafaei.sadr@gmail.com; 7Iranian Center of Neurological Research, Neuroscience Institute, Tehran University of Medical Sciences, Tehran 14155-6559, Iran; Ghasem.farahmand89@gmail.com; 8Research Institute of Neurointervention, Paracelsus Medical University, 5020 Salzburg, Austria; 9Department of Clinical Neurological Sciences, Western University, London, ON N6A 3K7, Canada; azarpazhoohr@gmail.com; 10Steele Institute of Health and Innovation, Geisinger Health System, Danville, PA 17822, USA; dmisra@geisinger.edu; 11Department of Molecular and Functional Genomics, Geisinger Health System, Danville, PA 17822, USA; jli@geisinger.edu (J.L.); vabedi@geisinger.edu (V.A.); 12Biocomplexity Institute, Virginia Tech, Blacksburg, VA 24060, USA

**Keywords:** cerebrovascular disorders, stroke, SARS-CoV-2, COVID-19, cluster analysis, risk factors, comorbidity

## Abstract

Background. SARS-CoV-2 infected patients are suggested to have a higher incidence of thrombotic events such as acute ischemic strokes (AIS). This study aimed at exploring vascular comorbidity patterns among SARS-CoV-2 infected patients with subsequent stroke. We also investigated whether the comorbidities and their frequencies under each subclass of TOAST criteria were similar to the AIS population studies prior to the pandemic. Methods. This is a report from the Multinational COVID-19 Stroke Study Group. We present an original dataset of SASR-CoV-2 infected patients who had a subsequent stroke recorded through our multicenter prospective study. In addition, we built a dataset of previously reported patients by conducting a systematic literature review. We demonstrated distinct subgroups by clinical risk scoring models and unsupervised machine learning algorithms, including hierarchical K-Means (ML-K) and Spectral clustering (ML-S). Results. This study included 323 AIS patients from 71 centers in 17 countries from the original dataset and 145 patients reported in the literature. The unsupervised clustering methods suggest a distinct cohort of patients (ML-K: 36% and ML-S: 42%) with no or few comorbidities. These patients were more than 6 years younger than other subgroups and more likely were men (ML-K: 59% and ML-S: 60%). The majority of patients in this subgroup suffered from an embolic-appearing stroke on imaging (ML-K: 83% and ML-S: 85%) and had about 50% risk of large vessel occlusions (ML-K: 50% and ML-S: 53%). In addition, there were two cohorts of patients with large-artery atherosclerosis (ML-K: 30% and ML-S: 43% of patients) and cardioembolic strokes (ML-K: 34% and ML-S: 15%) with consistent comorbidity and imaging patterns. Binominal logistic regression demonstrated that ischemic heart disease (odds ratio (OR), 4.9; 95% confidence interval (CI), 1.6–14.7), atrial fibrillation (OR, 14.0; 95% CI, 4.8–40.8), and active neoplasm (OR, 7.1; 95% CI, 1.4–36.2) were associated with cardioembolic stroke. Conclusions. Although a cohort of young and healthy men with cardioembolic and large vessel occlusions can be distinguished using both clinical sub-grouping and unsupervised clustering, stroke in other patients may be explained based on the existing comorbidities.

## 1. Introduction

Since the emergence of the Coronavirus Disease 2019 (COVID-19) pandemic, many cerebrovascular events have been reported among patients with SARS-CoV-2 infection. Some reports have highlighted strokes in critically ill and older patients with a higher number of comorbidities, while others have suggested a higher risk in younger and healthy individuals [1,2,3,4,5]. Studies have suggested that stroke patients with SARS-CoV-2 present with multiple cerebral infarcts [2,4,6], systemic coagulopathies [7], uncommon thrombotic events such as aortic [8] or common carotid artery thrombosis [9], and simultaneous arterial and venous thrombus formation [10].

Considering the hypercoagulable state as one of the main etiologies of stroke among the SARS-CoV-2 infected patients, we would expect a similar increased rate for cardiovascular thrombotic events and acute coronary syndrome after the pandemic. However, higher acute coronary syndrome case fatality rate and other adverse outcomes among cardiac patients compared with the pre-pandemic era have been attributed to public fear and reluctance to call for medical aid and increased pre-hospital delay. A dramatic decline in the guideline-indicated care, hospitalization rate, and revascularization procedures are other possible factors attributing to adverse outcomes in patients with acute coronary syndrome [11,12,13,14,15]. Studies have failed to show any difference among cardiovascular patients in terms of age, sex, comorbidities, clinical presentation, and diagnosis pre- and post-pandemic era [14,16]. Similarly, a higher rate of coronary stent thrombosis in comparison with the pre-pandemic era [17,18] was reported among the patients with multiple comorbidities (about 44% with at least four vascular risk factors) and a median age of 65 years [18]. Acute myocardial injury (defined as a substantial increase in cardiac troponin level) is associated with the underlying cardiac pathology in the majority of the SARS-CoV-2 infected patients [19] rather than a thrombotic event.

The first report from our Multinational COVID-19 Stroke Study Group and recent meta-analyses on reported infected patients presented a stroke incidence rate of 0.5–1.4% [20,21,22]. The odds of stroke after SARS-CoV-2 may not be greater than in non-infected patients [23]. In addition, meta-analyses of the reported patients presented that SARS-CoV-2 infected patients who experienced a stroke had a mean age of over 65 years, carried a load of comorbidities, and were affected by more severe infections [21,22]. Thereby, in some patients, stroke may be a coincidence or an indirect consequence of critical illness [24,25] and not a direct complication of the SARS-CoV-2 infection. As an example, there is an increased risk of ischemic stroke (odds ratio (OR) > 28) and hemorrhagic stroke (OR > 12) within two weeks of sepsis [26]. This might be due to new-onset atrial fibrillation (6%) that put the patient at risk of in-hospital stroke (2.6%) [24].

Understanding the population at risk for having a stroke after SARS-CoV-2 infection can promote timely diagnosis and proper management of these patients.

We designed this study to explore the pattern of traditional vascular risk factors and stroke etiology among stroke patients with prior SARS-CoV-2 infection. We leveraged unsupervised hierarchical and spectral model-based clustering in addition to clinical risk scoring models to decipher patterns of comorbidity among stroke patients with prior SARS-CoV-2 infection. We further expanded our analysis to corroborate whether the comorbidities under each subclass of TOAST (the Trial of Org 10172 in Acute Stroke Treatment [27]) were similar to the AIS population studies prior to the pandemic.

## 2. Methods

This report presents a multicenter prospective and observational study from our Multinational COVID-19 Stroke Study Group [20] and a cohort of patients extracted from the literature.

### 2.1. Original Dataset

Collaborators from 71 centers of 17 countries (Brazil, Canada, Croatia, Egypt, France, Germany, Greece, Iran, Israel, Italy, Portugal, Republic of Korea, Singapore, Spain, Switzerland, Turkey, and the United States) reported data on their patients for this study. We included consecutive SARS-CoV-2 infected adult patients who had imaging confirmed subsequent acute ischemic stroke.

The study protocol, details of eligibility criteria, data elements, and neurological investigations have been previously published [20]. The demographics, vascular risk factors, and comorbidities—i.e., hypertension, diabetes, ischemic heart disease, atrial fibrillation, carotid stenosis, chronic kidney disease, congestive heart failure with cardiac ejection fraction <40%, active neoplasms, rheumatological diseases, smoking status, and history of transient ischemic attack (TIA) or stroke—were recorded for the stroke patients [28,29,30,31]. We also recorded the neurological examinations, the National Institute of Health Stroke Scale (NIHSS), TOAST [27] subclasses, presence of large-vessel occlusions (LVOs), and brain imaging findings.

The study protocol was designed at the Neuroscience Institute of Geisinger Health System, Pennsylvania, United States, and received approval by the Institutional Review Board of Geisinger Health System and participating institutions, as needed. The study was conducted and reported according to the Strengthening the Reporting of Observational Studies in Epidemiology (STROBE) [32], and Preferred Reporting Items for Systematic Reviews and Meta-Analyses (PRISMA) [33].

### 2.2. Systematic Literature Review

To compare our results with the available literature, we searched PubMed for reports of patients with subsequent stroke after SARS-CoV-2 infection. Different terms in addition to Medical Subject Headings (MeSH) were utilized to build the search protocol (Appendix A). The search was last updated on 15 October 2020, with no limitation to study design, language, or document type. The search was augmented by forward and backward citation tracking in PubMed and Google Scholar. We additionally searched medRxiv to track the documents ahead of publication and communicated with the corresponding authors to include them under peer review or in press studies prior to publication. Two reviewers (EK and SS) independently evaluated the titles/abstracts of the retrieved results and reviewed the full texts of candidate articles. Data available from the literature were extracted per the same datasheet as the data collected in our original multicenter case series when possible. The extracted data were further reviewed by two neurologists (G.F. and R.Z.).

### 2.3. Comorbidity-Based Subgrouping: Expert Opinion

The details of the subgroups are available in Appendix A. In the risk scoring models based on the EXpert opinion (EX), we considered the number of present stroke-related comorbidities—either All the 11 collected comorbidities (EX-A) as mentioned above, or eight Selected comorbidities (EX-S, excluding congestive heart failure, active neoplasm, and rheumatological disorders) [27,28,29,30]. We considered equal weights for all comorbidities. We divided the patients based on EX-A and EX-S scores into two subgroups (EX-A_2_ and EX-S_2_); Subgroup “a” included patients who had a history of zero or one stroke-related comorbidity, and subgroup “b” included the patients with >1 comorbidity. In addition, we divided the patients based on EX-A and EX-S scores into three subgroups (EX-A_3_ and EX-S_3_). In this second classification, subgroup “a” represented the patients without any known comorbidity, subgroup “b” with one or two comorbidities, and subgroup “c” included the patients with more than two comorbidities.

### 2.4. Comorbidity-Based Subgrouping: Unsupervised Modeling

We explore the probable similarities among the patients based on the presence of comorbidities in a data-driven approach. These patterns might have been remained hidden by clinical risk scores to the experts. For this purpose, we leveraged unsupervised algorithms and Machine Learning models (ML) (Appendix A). We applied hierarchical (complete linkage method) and K-means (Hartigan-Wong algorithm) clustering (ML-K models) to group the patients into 2 (ML-K_2_) and 3 (ML-K_3_) subgroups. We also used Spectral clustering [34] (ML-S models) and clustered the patients into two (ML-S_2_) and three subgroups (ML-S_3_). Appendix A present the clustering of the patients into four and five subgroups. Patients from the original dataset and literature review were clustered independently.

We used the contingency matrix (also known as a contingency table) [35] to demonstrate the subgroups of each model versus other models. The average similarity of the models in clustering the patients was calculated as $Sim=\;\frac{\sum_{1}^{i}Maximum\;Value\;in\;Column\;i}{\sum_{1}^{k}Value\;in\;Cell\;k};$ where *i* is the number of columns and *k* is the total number of cells in the contingency matrix. Similarities among the models were considered as mild (50–65%), moderate (65–80%), and strong (80–100%). The packages *stat* [36] and *gplots* [37] in R version 3.6.3, and the scikit-learn package [38] in Python version 3.7 were used.

### 2.5. Statistical Analysis

We used descriptive statistics to summarize the data. Demographic data, comorbidities, laboratory findings, and neurological investigations were reported as medians (interquartile range (IQR)) and mean (standard deviations (SD)). Categorical variables were reported as absolute frequencies and percentages. The comparison between categorical variables was conducted with the Pearson chi-square test, while the differences among continuous variables were assessed by an independent *t*-test. We explored the association of comorbidities with each subclass of TOAST criteria by binary logistic regression. A *p*-value < 0.05 was considered significant in all analyses.

## 3. Results

### 3.1. Patients Characteristics

This study included 323 AIS patients from our original prospective multicenter case series, with a mean age of 67 ± 15 years and 60% men (Appendix A). The most prevalent comorbidities were hypertension (63%), diabetes (35%), and ischemic heart disease (24%). In addition, through our systematic review of the literature, we retrieved data from an additional 412 stroke patients (including dural sinus thrombosis) post-SARS-CoV-2 infection (Figure 1). The data from the 412 patients were extracted from 81 articles (in 18 countries). Among the 412 patients, individual-level data of 145 AIS patients were reported from 36 centers in nine countries. The mean age of the retrieved AIS patients was 63 ± 14 years, and 57% were men (Appendix A).

In comparison with our original multicenter dataset, patients reported in the literature were younger (mean age of 63 versus 67 years, *p* < 0.01), with a higher proportion of LVOs (83% versus 45%, *p* < 0.0001), and strokes of undetermined (38% versus 22%, *p* < 0.01) or other determined etiologies (31% versus 8%, *p* < 0.001). Although not statistically significant, reported patients in the literature had more severe strokes (median NIHSS of 15 versus nine, *p* = 0.11). Fewer patients of this cohort were reported to have had vascular risk factors; however, hypertension (55%), diabetes (37%), and atrial fibrillation (12%) were the most prevalent reported comorbidities among the patients from the published reports.

### 3.2. Clinical Risk Scoring Models Revealed a Large Cohort of Young Men with No Comorbidities Who Suffered from Large Vessel Occlusions (LVOs)

Among the 323 AIS patients from the original dataset, 65 (22%) patients reported no known comorbidities, and 115 (39%) had at most one known comorbidity (Table 1). Among the 117 patients from the literature review who had a completed comorbidity panel, 33 (28%) reported no known comorbidity, and 71 (61%) had at most one known comorbidity (Appendix A).

In both datasets, we identified a cohort of patients with no vascular risk factors with distinct features—subgroup “a” in all clinical risk scoring models; original dataset, EX-A_3_a: 22% and EX-S_3_a: 25% (Table 1); literature review, EX-A_3_a = EX-S_3_a: 28% (Appendix A). These cohorts included patients with (1) younger age (over 8 years in comparison with other subgroups of the original dataset), (2) male predominance (original dataset, EX-A_3_a: 55% and EX-S_3_a: 54%; literature review, EX-A_3_a = EX-S_3_a: 59%), and (3) a higher proportion of embolic-appearing imaging stroke pattern (original dataset, 82%; literature review dataset 67%). About half of patients in the original dataset had LVOs (EX-A_3_a: 48% and EX-S_3_a: 49%), as did the majority of patients reported in the literature (EX-A_3_a = EX-S_3_a: 80%). In comparison with patients who carried a high load of comorbidities (subgroup “c”), the cohorts of patients without comorbidities (subgroup “a”) had a longer length of hospital stay (original dataset EX-S_3_a, 16 days versus 11 days in EX-S_3_c, *p* = 0.03). Although not statistically significant, patients in the subgroup “a” also had less severe strokes (median NIHSS in the original dataset, eight versus 12 in subgroup “c”; median NIHSS in review dataset, six versus nine in subgroup “c”), but a higher chance of a need for mechanical ventilation (original dataset EX-A_3_a: 34% versus 28%, *p* = 0.39; EX-S_3_a: 37% versus 28%, *p* = 0.16).

### 3.3. Unsupervised Clustering Revealed Three Subgroups of Stroke Patients

In addition to clinical risk scoring, we used unsupervised algorithms to potentially identify hidden comorbidity patterns among AIS patients. There were strong similarities (Sim > 80%) among the models in grouping the patients, except two sets that were moderately similar (Appendix A). Clustering the patients from the original dataset (Table 2) demonstrated a subgroup of patients with no or few comorbidities—subgroup “a” in all ML models (ML-K_3_a: 36% and ML-S_3_a: 42% of patients, Table 2). The latter is similar to subgroup “a” in all EX models (22–46% of patients, Table 1). The patients in these groups were (1) mainly men (ML-K_3_a: 59% and ML-S_3_a: 60%), (2) more than six years younger than other subgroups, (3) had a higher risk of embolic-appearing stroke on imaging (ML-K_3_a: 83% and ML-S_3_a: 85%), and (4) had about 50% risk of LVOs (ML-K_3_a: 50% and ML-S_3_a: 53%). Patients in the second subgroup (ML-K_3_b: 30% and ML-S_3_b: 43%; similar to EX-A_3_b: 47% and EX-S_3_b: 50%) presented with a high proportion of hypertension, diabetes, chronic kidney disease, and smoking. These patients had a higher risk of large artery atherosclerosis (ML-K_3_b: 40%, and ML-S_3_b: 31%). The third subgroup (ML-K_3_c: 34% and ML-S_3_c: 15% similar to EX-A_3_c: 31% and EX-S_3_c: 25%) presented mostly with hypertension, diabetes, ischemic heart disease, atrial fibrillation, congestive heart failure, carotid stenosis, neoplasm, and smoking. The majority of these patients (ML-K_3_c: 34% and ML-S_3_c: 60%) had cardioembolic strokes based on TOAST and imaging patterns consistent with an embolic ischemic stroke.

Similar patterns were observed among patients reported in the literature (Appendix A). The first group (subgroup “a” in all models, 28–61%) included the patients with no or few comorbidities. These patients were more likely men (63–100%), with over 80% LVOs, about 65% strokes of undetermined or other determined etiologies, and over 60% embolic-appearing strokes. In the second subgroup identified by unsupervised clustering (ML-K_3_b: 41% and ML-S_3_b: 66%, similar to EX-A_3_b: 33% and EX-S_3_b: 33%), the majority of the patients presented with hypertension and diabetes. Strokes of undetermined (ML-K_3_b: 39% and ML-S_3_b: 33%) and other determined (ML-K_3_b: 33% and ML-S_3_b: 37%) etiologies were more prevalent in these subgroups. The third subgroup (ML-K_3_c: 16% and ML-S_3_c: 26%, similar to EX-A_3_c: 39% and EX-S_3_c: 39%) included the patients with hypertension, diabetes, ischemic heart disease, atrial fibrillation, smoking, and prior stroke or TIA. The majority of the patients in the third subgroup of the literature review dataset had strokes of undetermined (ML-K_3_c, 46% and ML-S_3_c, 50%) or other determined etiologies (ML-K_3_c: 27% and ML-S_3_c: 18%).

### 3.4. The TOAST Subtype Classification Was Consistent with the Patients’ Risk Profile

We observed significantly different proportions of hypertension, ischemic heart disease, atrial fibrillation, carotid stenosis, chronic kidney disease, and active neoplasms among subclasses of TOAST (Table 3). Binominal logistic regression models demonstrated that atrial fibrillation (OR: 0.2; 95% CI: 0.04–0.8) and carotid stenosis (OR: 6.9; 95% CI: 2.2–21.4) were associated with large-artery atherosclerosis; ischemic heart disease (OR: 4.9; 95% CI: 1.6–14.7), atrial fibrillation (OR: 14.0; 95% CI: 4.8–40.8), and active neoplasm (OR: 7.1; 95% CI: 1.4–36.2) with cardioembolic stroke; chronic kidney disease (OR: 6.23; 95% CI: 1.8–21.5) with small-vessel occlusion; and ischemic heart disease (OR: 0.1; 95% CI: 0.01–0.5), carotid stenosis (OR: 0.1; 95% CI: 0.01–0.8), and chronic kidney disease (OR: 0.2; 95% CI: 0.04–0.9) with strokes of other determined etiology.

Among the AIS patients reported in the literature, 120 patients had available TOAST criteria, 109 patients had available comorbidity panel, and 93 patients had data regarding both the TOAST criteria and the comorbidities. Because of the small sample size under each subgroup of TOAST, further analysis of the association of TOAST and comorbidities among these patients was not performed.

## 4. Discussion

The results of our study indicated that SARS-CoV-2 infection could cause AIS among a considerable number of young and majority male patients who did not have vascular risk factors. The majority of these young patients had embolic-appearing stroke on their neuroimaging. Stroke in older patients can be attributed to the existing vascular risk factors.

### 4.1. Unsupervised Clustering Identified Three Subgroups of SARS-CoV-2 Infected AIS Patients

Despite several reports of special features and probable underlying coagulopathy in AIS with prior SARS-CoV-2 infection [2,4,6,7,8,9,10], similar reports are lacking in the literature regarding acute coronary syndrome and cardiovascular thromboembolic events. The majority of adverse outcomes among patients with stroke [39,40] or acute coronary syndrome [11,12,13,14,15] were related to the declining trend in seeking urgent care, hospitalization, and receiving guideline indicated measures. On the other hand, the meta-analyses of AIS infected patients presented a mean age of over 65 years and a high load of comorbidities [21,22]. Thereby, there might be a specific group of AIS patients with prior SARS-CoV-2 infection that can be attributed to the virus, while the incidence of stroke among other patients, especially older patients, might be related to their vascular risk factors or critical illness. On this basis, we analyzed the data from our Multinational COVID-19 Stroke Study Group [20] and a dataset of reported patients in the literature. The two cohorts facilitated the identification of three main subgroups. The first group includes patients with no or very few comorbidities—EX-A_3_a, EX-S_3_a, ML-K_3_a, and ML-S_3_a. The majority of these patients are young men who had an embolic-appearing stroke. The second subgroup was distinguishable by having a high proportion of hypertension, diabetes, chronic kidney disease, and carotid stenosis, large-artery atherosclerosis origin of stroke, and embolic-appearing stroke on imaging—ML-K_3_b, ML-S_3_b, EX-A_3_b, and EX-S_3_b. The third group presented with hypertension, diabetes, ischemic heart disease, atrial fibrillation, congestive heart failure, smoking, and prior TIA or stroke—ML-K_3_c, ML-S_3_c, EX-A_3_c, and EX-S_3_c. The majority of the patients in the third group had cardioembolic strokes based on the TOAST classification and had a consistent imaging pattern. Subgroups of patients identified by clinical risk scoring and unsupervised clustering based on the comorbidity panels were similar in the original and literature review datasets. However, unlike the original dataset, the etiology of the stroke in the majority of patients in the second and third subgroups of the review datasets were reported as “strokes of undetermined etiology”. Overall, the identified pattern demonstrated by all models may indicate that AIS in only a subgroup of patients can be attributed to the SARS-CoV-2 infection (subgroup a in all models), while AIS in the second and third group of patients may be explained by the existing comorbidities.

### 4.2. Higher Proportion of AIS Showed Lack of Comorbidities among SARS-CoV-2 Infected Patients

Our study indicated a subgroup of patients with no known comorbidities among the SARS-CoV-2 infected patients (22.0%).The result of our systematic literature review on SARS-CoV-2 infected stroke patients reported from 36 centers in nine countries similarly demonstrated that 24% of the patients had no prior comorbidities. The proportion of the patients without known comorbidities was not available from large-scale studies on SARS-CoV-2 infected stroke patients reported from the UK [5] and the Global COVID-19 Stroke Registry [41]. However, a case series from New York presented that among 32 infected AIS patients, seven (22%) did not report hypertension, diabetes, dyslipidemia, coronary artery disease, congestive heart failure, atrial fibrillation, prior stroke or transient ischemic stroke, or active smoking [42]. A series of 22 AIS patients with SARS-CoV-2 infection from the US demonstrated that 12 out of 22 (54%) of the patients did not report any comorbidities (i.e., hypertension, congestive heart failure, chronic lung disease, chronic kidney disease, diabetes, or atrial fibrillation) [43]. In a report of six consecutive SARS-CoV-2 infected AIS patients from the UK, one patient (16%) had no prior medical history [44]. All of these patients had LVO strokes and elevated D-dimer levels. Similarly, among the five young patients in the US who had LVO stroke after SARS-CoV-2, 2 (40%) reported no prior comorbidities [1]. These findings may suggest that after SARS-CoV-2 infection, higher percentages of patients without comorbidities are having a stroke.

### 4.3. The Proportion of Comorbidities under Each Subclass of TOAST Is Similar to Population Studies Prior to the Pandemic

The second report from our Multinational COVID-19 Stroke Study Group [20] indicated a lower rate of small-vessel occlusion and lacunar infarcts and a higher risk of embolic-appearing stroke in patients with SARS-CoV-2 infection in comparison with population studies conducted prior to the pandemic. These findings were valid even after considering the geographical regions and countries’ health expenditure. The results of subgroup analyses and binary logistic regression in the current study presented that the comorbidity panel of the patients from the original dataset is consistent with the stroke subtypes. To see if the comorbidity panel of AIS patients infected with SARS-CoV-2 was consistent with the large-scale population studies, we further investigated the proportion of comorbidities under each subclass of TOAST (Table 3). We observed that in comparison with population studies, AIS patients infected with SARS-CoV-2 have an almost similar rate of comorbidities under each subclass of TOAST [45,46,47,48]. Among patients with large-artery atherosclerosis in our study, 54% had hypertension (versus 54–85%), 36% had diabetes (versus 13–32%), and 20% were smokers (versus 17–50%). Among patients with cardioembolism, hypertension was recorded in 76% (versus 59–86%), diabetes in 33% (versus 13–32%), ischemic heart disease in 46% (versus 20–32%), and atrial fibrillation in 50% (versus 79–86%). Similarly, patients with small-vessel occlusion had 59% hypertension (versus 54–58%), 35% diabetes (versus 12–35%), and 18% ischemic heart disease (versus 15–20%) [45,46,47,48]. The result of the literature review presented similar findings, although we recognized a selective report of patients with a lower comorbidity panel (Table 3). These findings suggest that the comorbidities under each stroke etiology are not highly different from the population studies prior to the pandemic, and we should still consider the possibility of bias in reporting the patients with SARS-CoV-2 infection and stroke before concluding the role of the virus as an absolute direct cause of stroke.

## 5. Study Limitations

To build up the database of SARS-CoV-2 infected patients with stroke, several attempts have been made in collaboration with multiple centers around the world. In addition, we reviewed all available reports to present a comprehensive overview of the topic. Despite this effort, these findings could largely be affected by selection and low sample size bias as well as bias due to incomplete diagnostic workups. In addition, we could not include dyslipidemia in the comorbidity list because data regarding lipid profile could not pass the quality control phase. For instance, some of the included patients were reported before comprehensive diagnostic tests, which may cause a bias in determining the subclasses of TOAST criteria. We also detected publication bias among the reported patients in the literature (significantly lower age, higher LVOs, more severe strokes, and strokes with undetermined and other determined etiologies). In addition, clustering the patients in this study is limited to the vascular risk factors, and we did not include the laboratory findings. Lastly, the unsupervised algorithms tend to be susceptible to the presence of outliers, especially when used for data with a small sample size.

## 6. Conclusions

Among patients with SARS-CoV-2 and acute ischemic stroke, there is a considerable number of young and majority male patients who did not report vascular risk factors. Therefore, young patients with SARS-CoV-2 infection should be monitors for the sign and symptoms of vascular events, including ischemic stroke. It is reasonable to ensure that these patients and their families are aware of early signs of stroke (BE-FAST) [49]. Stroke in other patients can be attributed to the existing comorbidity panel. We also observed that the proportions of comorbidities under each subclass of TOAST criteria were not different from the population studies prior to the SARS-CoV-2 pandemic.

## Figures and Tables

**Figure 1 jcm-10-00931-f001:**
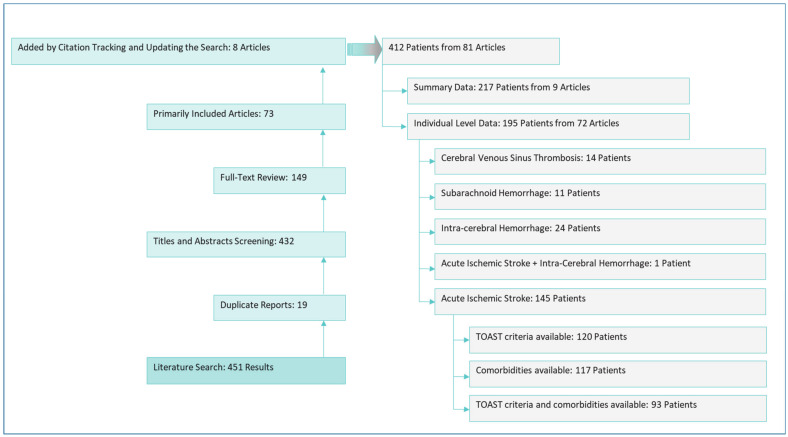
The process of literature review and main results.

**Table 1 jcm-10-00931-t001:** Characteristics of the patients grouped by clinical risk scoring models.

Parameters	Clinical Risk Scoring													
	EX-A_2_ (All Comorbidities)			EX-S_2_ (Selected Comorbidities)			EX-A_3_ (All Comorbidities)				EX-S_3_ (Selected Comorbidities)			
	a *n* = 115 (38.9%)	b *n* = 181 (61.1%)	*p*-Value	a *n* = 137 (46.3%)	b *n* = 159 (53.7%)	*p*-Value	a *n* = 65 (22.0%)	b *n* = 140 (47.3%)	c *n* = 91 (30.7%)	*p*-Value	a *n* = 74 (25.0%)	b *n* = 147 (49.7%)	c *n* = 75 (25.3%)	*p*-Value
**Age; Mean (SD); Years**	61 ± 18	69 ± 14	<0.001	62 ± 17	69 ± 14	<0.001	60 ± 18	68 ± 14	70 ± 14	<0.001	59 ± 18	69 ± 13	71 ± 13	<0.001
S**ex; Male; *n* (%)**	66 (57.4)	113 (62.4)	0.29	72 (56.3)	107 (63.7)	0.61	36 (55.4)	87 (62.1)	56 (61.5)	0.63	36 (54.5)	98 (64.9)	45 (57.0)	0.46
**Large Vessel Occlusion; *n* (%)**	43 (43.9)	76 (44.4)	0.93	49 (41.2)	70 (46.7)	0.37	26 (48.1)	50 (39.4)	43 (50.0)	0.26	31 (49.2)	52 (39.1)	36 (50.7)	0.20
**Intravenous Thrombolysis; *n* (%)**	13 (7.4)	26 (12.4)	0.11	16 (8.0)	23 (12.3)	0.17	6 (9.2)	14 (10.0)	19 (20.9)	0.03	7 (9.5)	17 (9.5)	15 (20.0)	0.12
**Mechanical Thrombectomy; *n* (%)**	9 (5.1)	15 (7.1)	0.41	10 (5.0)	14 (7.5)	0.32	5 (7.7)	9 (6.4)	10 (11.0)	0.46	6 (8.1)	9 (6.1)	9 (12.0)	0.32
**National Institutes of Health Stroke Scale (NIHSS); Median (IQR)**	11.0 ± 9.0	12.0 ± 9.0	0.95	11.0 ± 9.0	12.0 ± 8.0	0.87	8 (4–22)	9 (4–16)	12 (6–20)	0.18	9 (4–22)	8 (4–16)	12(6–19)	0.21
**TOAST Criteria**														
** 2003 Large-Artery Atherosclerosis; *n* (%)**	21 (30.4)	35 (34.7)	<0.001	21 (27.3)	35 (37.6)	<0.001	16 (43.2)	21 (26.9)	19 (34.5)	<0.001	16 (42.1)	24 (27.3)	16 (36.4)	<0.001
** Cardio-Embolism; *n* (%)**	10 (14.5)	36 (35.6)		12 (15.6)	34 (36.6)		5 (13.5)	13 (16.7)	28 (50.9)		5 (13.2)	20 (22.7)	21 (47.7)	
**Small-Vessel Occlusion; *n* (%)**	7 (10.1)	10 (9.9)		8 (10.4)	9 (9.7)		1 (2.7)	12 (15.4)	4 (7.3)		1 (2.6)	12 (13.6)	4 (9.1)	
** Stroke of Other Determined Etiology; *n* (%)**	11 (15.9)	2 (2.0)		11 (14.3)	2 (2.2)		6 (16.2)	7 (9.0)	0.0 (0.0)		7 (18.4)	6 (6.8)	0 (0.0)	
** Stroke of Undetermined Etiology; *n* (%)**	20 (29.0)	18 (17.8)		25 (32.5)	13 (14.0)		9 (24.3)	25 (32.1)	4 (7.3)		9 (23.7)	25 (29.5)	3 (6.5)	
**Imaging Patterns**														
** Embolic-Appearing; *n* (%)**	76 (83.5)	195 (92.9)	0.43	189 (95.0)	173 (92.5)	0.56	41 (82.0)	97 (79.5)	67 (80.7)	0.31	42(82.4)	106 (80.3)	57 (79.2)	0.72
** Lacune; *n* (%)**	10 (11.0)	16 (9.8)		14 (12.5)	12 (8.4)		4 (8.0)	17 (13.9)	5 (6.0)		4 (7.8)	17(12.9)	5 (6.9)	
** Borderzone; *n* (%)**	5 (5.5)	18 (11.0)		9 (8.0)	14 (9.8)		5 (10.0)	8 (6.6)	10 (12.0)		5 (9.8)	9 (6.8)	9 (12.5)	
** Vasculitis Pattern; *n* (%)**	0 (0.0)	1 (0.6)		0 (0.0)	1 (0.7)		0 (0.0)	0 (0.0)	1 (1.2)		0 (0.0)	0 (0.0)	1 (1.4)	
**Interval Between Infection Onset to Stroke; Median (IQR); Days**	7.0 ± 8.0	5.0 ± 6.0	0.07	7.0 ± 7.0	5.0 ± 6.0	0.15	7.0 ± 8.0	5.0 ± 6.0	5.0 ± 7.0	0.19	7.0 ± 7.0	5.0 ± 6.0	6.0 ± 7.0	0.27
**Mechanical Ventilation; *n* (%)**	22 (33.8)	63 (27.3)	0.30	27 (36.5)	58 (26.1)	0.09	22 (33.8)	38 (27.1)	25 (27.5)	0.39	27 (36.5)	37 (25.2)	21 (28.0)	0.16
**Disposition**														
** Discharged Home; *n* (%)**	66 (42.0)	77 (36.7)	0.46	75 (41.7)	68 (36.4)	0.44	31 (50.8)	60 (43.2)	36 (39.6)	0.39	32 (51.6)	62 (41.3)	33 (41.8)	0.16
** In Hospital Mortality; *n* (%)**	45 (28.7)	72 (34.3)		52 (28.9)	65 (34.8)		14 (23.0)	33 (23.7)	30 (33.0)		14 (22.6)	35 (23.3)	28 (35.4))	
** Still in Hospital/Subacute Care; *n* (%)**	46 (29.3)	61 (29.0)		53 (29.4)	54 (28.9)		16 (26.2)	46 (33.1)	25 (27.5)		16 (25.8)	53 (35.3)	18 922.8)	
**Length of Hospital Stay; Median (IQR); Days**	14.0 ± 15.0	11.0 ± 11.0	0.46	16.0 ± 17.0	11.0 ± 9.0	0.04	14.0 ± 15.0	12.0 ± 12.0	10.0 ± 8.0	0.28	16.0 ± 17.0	11.0 ± 9.0	11.0 ± 8.0	0.03
**Comorbidities**														
** Hypertension; *n* (%)**	22 (19.1)	158 (87.3)	<0.001	30 (23.4)	150 (89.3)	<0.001	0 (0.0)	95 (67.9)	85 (93.4)	<0.001	0 (0.0)	104 (68.9)	76 (96.2)	<0.001
** Diabetes Mellitus; *n* (%)**	5 (4.3)	93 (51.4)	<0.001	8 (6.3)	90 (53.6)	<0.001	0 (0.0)	37 (26.4)	61 (67.0)	<0.001	0 (0.0)	41 (27.2)	57 (72.2)	<0.001
** Ischemic Heart Disease; *n* (%)**	4 (3.5)	68 (37.6)	<0.001	4 (3.1)	68 (40.5)	<0.001	0 (0.0)	19 (13.6)	53 (58.2)	<0.001	0 (0.0)	24 (15.9)	48 (60.8)	<0.001
** Atrial Fibrillation; *n* (%)**	4 (3.5)	38 (21.0)	<0.001	6 (4.7)	36 (21.4)	<0.001	0 (0.0)	12 (8.6)	30 (33.0)	<0.001	0 (0.0)	15 (9.9)	27 (34.2)	<0.001
** Carotid Stenosis; *n* (%)**	1 (0.9)	37 (20.4)	<0.001	1 (0.8)	37 (22.0)	<0.001	0 (0.0)	11 (7.9)	27 (29.7)	<0.001	0 (0.0)	11 (7.3)	27 (34.2)	<0.001
** Chronic Kidney Disease; *n* (%)**	9 (7.8)	32 (17.7)	0.02	9 (7.0)	32 (19.0)	0.003	0 (0.0)	23 (16.4)	18 (19.8)	<0.001	0 (0.0)	24 (15.9)	17 (21.5)	<0.001
** Cardiac Ejection Fraction <40%; *n* (%)**	1 (0.9)	23 (12.7)	<0.001	7 (5.5)	17 (10.1)	0.15	0 (0.0)	6 (4.3)	18 (19.8)	<0.001	0 (0.0)	10 (6.6)	13 (16.5)	0.003
** Active Neoplasm; *n* (%)**	0 (0.0)	21 (11.6)	<0.001	6 (4.7)	15 (8.9)	0.16	0 (0.0)	5 (3.6)	16 (17.6)	<0.001	0 (0.0)	12 (7.9)	9 (11.4)	0.02
** Rheumatological Disease; *n* (%)**	0 (0.0)	5 (2.8)	0.07	3 (2.3)	2 (1.2)	0.45	0 (0.0)	2 (1.4)	3 (3.3)	0.27	0 (0.0)	4 (2.6)	1 (1.3)	0.35
** Prior Stroke or Transient Ischemic Attack; *n* (%)**	1 (0.9)	4 (2.2)	0.38	1 (0.8)	4 (2.4)	0.29	0 (0.0)	2 (1.4)	3 (3.3)	0.27	0 (0.0)	2 (1.3)	3 (3.8)	0.18
** Smoking; *n* (%)**	3 (2.6)	45 (24.9)	<0.001	3 (2.3)	45 (26.8)	<0.001	0 (0.0)	18 (12.9)	30 (33.0)	<0.001	0 (0.0)	19 (12.6)	29 (36.7)	<0.001

EX-A_2_: clinical risk-scoring (expert opinion) model including all comorbidities; a, 0–1 comorbidity; b, >1 comorbidity; EX-S_2_: clinical risk-scoring model including selected comorbidities; a, 0–1 comorbidity; b, >1 comorbidity; EX-A_3_: clinical risk scoring model including all comorbidities; a, 0 comorbidity; b, 1–2 comorbidities, c, >2 comorbidities; EX-S_3_: clinical risk scoring model including selected comorbidities; a, 0 comorbidities; b, 1–2 comorbidities, c, >2 comorbidities. Due to missingness, we provided the valid percentages in this table.

**Table 2 jcm-10-00931-t002:** Characteristics of the patients clustered with unsupervised machine learning algorithms.

Parameters	Unsupervised Machine Learning Models													
	ML-K_2_ (K-Mean)			ML-S_2_ (Spectral)			ML-K_3_ (K-Mean)				ML-S_3_ (Spectral)			
	a *n* = 112 (38.4%)	b *n* = 180 (61.6%)	*p*-Value	a *n* = 173 (60.3%)	b *n* = 114 (39.7%)	*p*-Value	a *n* = 106 (36.3%)	b *n* = 87 (29.8%)	c *n* = 99 (33.9%)	*p*-Value	a *n* = 120 (41.8%)	b *n* = 123 (42.9%)	c *n* = 44 (15.3%)	*p*-Value
**Age; Mean (SD); Years**	62 ± 17	70 ± 13	<0.001	66 ± 17	68 ± 13	0.02	62 ± 17	68 ± 13	72 ± 13	<0.001	63 ± 17	70 ± 14	70 ± 14	<0.001
**Sex; Male; *n* (%)**	67 (59.8)	110 (61.1)	0.08	107 (61.8)	66 (57.4)	0.05	63 (59.4)	51 (58.6)	63 (63.6)	0.75	72 (60.0)	77 (62.6)	24 (53.3)	<0.001
**Large Vessel Occlusion; *n* (%)**	46 (48.4)	73 (42.7)	0.37	64 (42.1)	54 (50.0)	0.21	46 (49.5)	36 (44.4)	38 (40.4)	0.47	55 (53.4)	41 (35.3)	22 (53.7)	0.64
**Intravenous Thrombolysis; *n* (%)**	16 (14.3)	23 (12.8)	0.71	17 (9.8)	23 (17.5)	0.06	14 (13.2)	15 (17.2)	10 (10.1)	0.36	15 (12.5)	10 (8.1)	12 (27.3)	0.01
**Mechanical Thrombectomy; *n* (%)**	10 (8.9)	14 (7.8)	0.73	12 (6.9)	12 (10.5)	0.28	10 (9.4)	8 (9.2)	6 (6.1)	0.63	13 (10.8)	7 (5.7)	4 (9.1)	0.34
**National Institutes of Health Stroke Scale (NIHSS); Median (IQR**)	12.0 ± 9.0	11.0 ± 8.0	0.52	11.0 ± 8.0	13.0 ± 8.0	0.11	10 (5–19)	12 (6–18)	8 (4–16)	0.28	11 (5–19)	8 (4.16)	13 (7–20)	0.03
**TOAST Criteria**														
** Large-Artery Atherosclerosis; *n* (%)**	25 (38.5)	31 (29.8)	0.03	32 (31.1)	24 (35.8)	0.002	23 (36.5)	19 (40.4)	14 (23.7)	0.08	27 (38.6)	23 (30.7)	6 (24.0)	0.003
** Cardio-Embolism; *n* (%)**	11 (16.9)	35 (33.7)		19 (18.4)	27 (40.3)		11 (17.5)	15 (31.9)	20 (33.9)		14 (20.0)	17 (22.7)	15 (60.0)	
** Small-Vessel Occlusion; *n* (%)**	7 (10.8)	9 (8.7)		11 (10.7)	6 (9.0)		7 (11.1)	4 (8.5)	5 (8.5)		7 (10.0)	8 (10.7)	2 (8.0)	
** Stroke of Other Determined Etiology; *n* (%)**	9 (13.8)	4 (3.8)		12 (11.7)	1 (1.5)		9 (14.3)	1 (2.1)	3 (5.1)		9 (12.9)	4 (5.3)	0 (0.0)	
** Stroke of Undetermined Etiology; *n* (%)**	13 (20.0)	25 (24.0)		29 (28.2)	9 (13.4)		13 (20.6)	8 (17.0)	17 (28.8)		13 (18.6)	23 (30.7)	2 (8.0)	
**Imaging Patterns**														
** Embolic-Appearing; *n* (%)**	74 (83.1)	131 (79.4)	0.44	115 (79.3)	84 (80.8)	0.38	71 (82.6)	59 (77.6)	75 (81.5)	0.49	83 (84.7)	81 (73.6)	35 (85.4)	0.22
** Lacune; *n* (%)**	10 (11.2)	15 (9.1)		18 (12.4)	8 (7.7)		10 (11.6)	6 (7.9)	9 (9.8)		10 (10.2)	12 (10.9)	4 (9.8)	
** Borderzone; *n* (%)**	5 (5.6)	18 (10.9)		12 (8.3)	11 (10.6)		5 (5.8)	10 (13.2)	8 (8.7)		5 (5.1)	16 (14.5)	2 (4.9)	
** Vasculitis Pattern; *n* (%)**	0 (0.0)	1 (0.6)		0 (0.0)	1 (1.0)		0 (0.0)	1 (1.3)	0 (0.0)		0 (0.0)	1 (0.9)	0 (0.0)	
**Interval Between Infection Onset to Stroke; Median (IQR); Days**	6.0 ± 7.0	5.0 ± 6.0	0.19	6.0 ± 7.0	5.0 ± 7.0	0.38	7.0 ± 7.0	5.0 ± 7.0	5.0 ± 6.0	0.28	6.0 ± 7.0	5.0 ± 6.0	6.0 ± 8.0	0.37
**Mechanical Ventilation; *n* (%)**	36 (32.1)	47 (26.1)	0.27	51 (29.5)	32 (28.1)	0.80	34 (32.1)	24 (27.6)	25 (25.3)	0.55	39 (32.5)	31 (25.2)	13 (25.9)	0.27
**Disposition**														
** Discharged Home; *n* (%)**	53 (48.6)	72 (40.2)	0.36	81 (47.4)	43 (37.7)	0.27	50 (48.5)	38 (43.7)	37 (37.8)	0.57	56 (47.1)	51 (41.8)	17 (38.6)	0.27
** In Hospital Mortality; *n* (%)**	27 (24.8)	49 (27.4)		41 (24.0)	33 (28.9)		24 (23.3)	25 (28.7)	27 (27.6)		27 (22.7)	30 (24.6)	17 (38.6)	
** Still in Hospital/Subacute Care; *n* (%)**	29 (26.6)	58 (32.4)		49 (28.7)	38 (33.3)		29 (28.2)	24 (27.6)	34 (34.7)		36 (30.3)	41 (33.6)	10 (22.7)	
**Length of Hospital Stay; Median (IQR); Days**	14.0 ± 15.0	11 ± 9.0	0.14	13.0 ± 14.0	11.0 ± 9.0	0.23	14.0 ± 15.0	12.0 ± 9.0	10.0 ± 8.0	0.11	13.0 ± 15.0	12.0 ± 9.0	10.0 ± 7.0	0.56
**Comorbidities**														
** Hypertension; *n* (%)**	0 (0.0)	179 (99.4)	<0.001	65(37.6)	109 (94.8)	<0.001	0 (0.0)	80 (92.0)	99 (100.0)	<0.001	13 (10.8)	121 (98.4)	40 (88.9)	<0.001
** Diabetes Mellitus; *n* (%)**	16 (14.3)	81 (45.0)	<0.001	13(7.5)	83 (72.2)	<0.001	10 (9.4)	87 (100.0)	0 (0.0)	<0.001	11 (9.2)	51 (41.5)	34 (75.6)	<0.001
** Ischemic Heart Disease; *n* (%)**	16 (14.3)	55 (30.6)	0.002	13(7.5)	58 (50.4)	<0.001	10 (9.4)	36 (41.4)	25 (25.3)	<0.001	26 (21.7)	1 (0.8)	44 (97.8)	<0.001
** Atrial Fibrillation; *n* (%)**	10 (8.9)	31 (17.2)	0.05	14(8.1)	28 (24.3)	<0.001	9 (8.5)	14 (16.1)	18 (18.2)	0.11	9 (7.5)	16 (13.0)	17 (37.8)	<0.001
** Carotid Stenosis; *n* (%)**	4 (3.6)	34 (18.9)	<0.001	10(5.8)	27 (23.5)	<0.001	4 (3.8)	21 (24.1)	13 (13.1)	<0.001	5 (4.2)	15 (12.2)	17 (37.8)	<0.001
** Chronic Kidney Disease; *n* (%)**	14 (12.5)	27 (15.0)	0.55	28(16.2)	13 (11.3)	0.25	14 (13.2)	10 (11.5)	17 (17.2)	0.51	12 (10.0)	26 (21.1)	3 (6.7)	0.01
** Cardiac Ejection Fraction <40%; *n* (%)**	2 (1.8)	22 (12.2)	0.002	7(4.0)	17 (14.8)	<0.001	2 (1.9)	10 (11.5)	12 (12.1)	0.01	3 (2.5)	13 (10.6)	8 (17.8)	0.003
** Active Neoplasm; *n* (%)**	6 (5.4)	15 (8.3)	0.34	7 (4.0)	14 (12.2)	0.009	4 (3.8)	9 (10.3)	8 (8.1)	0.19	6 (5.0)	5 (4.1)	10 (22.2)	<0.001
** Rheumatological Disease; *n* (%)**	1 (0.9)	4 (2.2)	0.39	4 (2.3)	1 (0.9)	0.36	1 (0.9)	1 (1.1)	3 (3.0)	0.46	1 (0.8)	4 (3.3)	0 (0.0)	0.22
** Prior Stroke or Transient Ischemic Attack; *n* (%)**	2 (1.8)	3 (1.7)	0.94	2 (1.2)	3 (2.6)	0.36	1 (0.9)	3 (3.4)	1 (1.0)	0.33	1 (0.8)	2 (1.6)	2 (4.4)	0.28
** Smoking; *n* (%)**	7 (6.3)	41 (22.8)	<0.001	19 (11.0)	27 (23.5)	0.005	7 (6.6)	17 (19.5)	24 (24.2)	0.002	10 (8.3)	24 (19.5)	12 (26.7)	0.006

ML-K_2_: machine learning model using K-mean, dividing the patients into two subgroups; ML-S_2_: machine learning model using spectral, dividing the patients into two subgroups; ML-K_3_: machine learning model using K-mean, dividing the patients into three subgroups; ML-S_3_: machine learning model using spectral, dividing the patients into three subgroups. Please note a, b, and c in this table are not based on the number of comorbidities and just indicated a distinct subgroup detected by unsupervised algorithms. Due to missingness, we provided the valid percentages in this table.

**Table 3 jcm-10-00931-t003:** The proportion of comorbidities under each subgroup of TOAST in original dataset and literature review dataset. Due to missingness, the valid percentages are reported in this table.

Parameter	Original Data from Multicenter Study						Literature Review					
	Large Artery Atherosclerosis *n* = 56 (32.9%)	Cardio-Embolic *n* = 46 (27.1%)	Small Artery Occlusion *n* = 17 (10.0%)	Other Determined Etiologies *n* = 13 (7.6%)	Undetermined Etiology *n* = 38 (22.4%)	*p*-Value	Large Artery Atherosclerosis *n* = 12 (10.0%)	Cardio-Embolic *n* = 17 (14.2%)	Small Artery Occlusion *n* = 8 (6.7%)	Other Determined Etiologies *n* = 37 (30.8%)	Undetermined Etiology *n* = 46 (38.3%)	*p*-Value
**Hypertension *n* (%)**	30 (53.6)	35 (76.1)	10 (58.8)	4 (30.8)	25 (65.8)	0.025	6 (66.7)	7 (50.0)	2 (33.3)	15 (48.4)	19 (54.3)	0.762
**Diabetes Mellitus *n* (%)**	20 (35.7)	15 (32.6)	6 (35.3)	1 (7.7)	12 (31.6)	0.407	3 (33.3)	1 (7.1)	2 (33.3)	12 (38.7)	17 (48.6)	0.112
**Ischemic Heart Disease *n* (%)**	11 (19.6)	21 (45.7)	3 (17.6)	1 (7.7)	2 (5.3)	<0.001	3 (33.3)	1 (7.1)	0 (0.0)	1 (3.2)	3 (8.6)	0.063
**Atrial Fibrillation *n* (%)**	2 (3.6)	23 (50.0)	4 (23.5)	1 (7.7)	1 (2.6)	<0.001	1 (11.1)	2 (14.3)	0 (0.0)	3 (9.7)	7 (20.0)	0.625
**Carotid stenosis *n* (%)**	16 (28.6)	6 (13.0)	1 (5.9)	0 (0)	2 (5.3)	0.005	0 (0.0)	0 (0.0)	0 (0.0)	1 (3.2)	1 (2.9)	0.923
**Chronic Kidney Disease *n* (%)**	8 (14.3)	3 (6.5)	6 (35.3)	1 (7.7)	3 (7.9)	0.028	1 (11.1)	0 (0.0)	0 (0.0)	1 (3.2)	0 (0.0)	0.296
**Congestive Heart Failure with Cardiac Ejection Fraction < 40% *n* (%)**	5 (8.9)	8 (17.4)	1 (5.9)	1 (7.7)	5 (13.2)	0.612	0 (0.0)	0 (0.0)	0 (0.0)	0 (0.0)	1 (2.90)	0.785
**Active Neoplasm *n* (%)**	2 (3.6)	9 (19.6)	1 (5.9)	0 (0)	0 (0)	0.003	0 (0.0)	0 (0.0)	0 (0.0)	0 (0.0)	0 (0.0)	*
**Rheumatological Disease *n* (%)**	0 (0)	3 (6.5)	1 (5.9)	0 (0)	1 (2.6)	0.321	0 (0.0)	0 (0.0)	0 (0.0)	0 (0.0)	0 (0.0)	*
**Previous stroke/Transient Ischemic Attack *n* (%)**	0 (0)	0 (0)	0 (0)	0 (0)	1 (2.6)	0.479	2 (22.2)	0 (0.0)	0 (0.0)	2 (6.5)	3 (8.6)	0.315
**Current Smoker *n* (%)**	11 (19.6)	5 (10.9)	2 (11.8)	0 (0)	4 (10.5)	0.336	1 (11.1)	2 (14.3)	1 (16.7)	2 (6.3)	3 (8.6)	0.878

* Due to missingness, this value could not be computed. We provided the valid percentages in this table.

## Data Availability

The data presented in this study are available in the manuscript and supplemental materials. Additional data are available on request from the corresponding author. The data are not publicly available due to health information privacy.

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
