# Peer review of "SARS-CoV-2 Is a Culprit for Some, but Not All Acute Ischemic Strokes: A Report from the Multinational COVID-19 Stroke Study Group"

_jcm, 2021, doi:10.3390/jcm10050931_

Round 1
Reviewer 1 Report
This manuscript reports results of a multinational study conducted by the COVID19 Stroke Study Group to explore possible link between SARS-CoV-2 infection and AIS (Acute Ischemic Disease).
The manuscript is well written, the methodology and the results are extensively described with a logical flow.
Methods described in the manuscript look appropriate , results are clearly described and discussion has a scientific sounding.
I suggest to publish the manuscript in the present form.
Author Response
Reviewer One:
Comments and Suggestions for Authors
This manuscript reports results of a multinational study conducted by the COVID19 Stroke Study Group to explore possible link between SARS-CoV-2 infection and AIS (Acute Ischemic Disease).
The manuscript is well written, the methodology and the results are extensively described with a logical flow. Methods described in the manuscript look appropriate, results are clearly described and discussion has a scientific sounding. I suggest to publish the manuscript in the present form.
We appreciate the time and effort of reviewer one in reviewing our manuscript. The authors are delighted that the manuscript meets the quality considered by reviewer one.
Reviewer 2 Report
Major
It is presumably considered that the association between stroke and COVID-19 is probably multifactorial including an amalgamation of traditional vascular risk factors, proinflammatory and a prothrombotic state. Therefore, real-world collected data is required to reveal the mechanism and relationship between SARS-CoV-2 infection and stroke. In this point, the current study is very informative for many physicians and neurologists. The authors examined 323 acute ischemic strokes (AIS) patients from multicenter in 17 countries from the original dataset and 145 patients reported in the literature. Majority of patients suffered from an embolic-appearing stroke on imaging, and had about 50% risk of large vessel occlusions. The binominal logistic regression revealed that ischemic heart disease (OR, 4.9), atrial fibrillation (OR, 14.0), and active neoplasm (OR, 7.1) are associated with cardio embolic stroke. Since the hypercoagulable state is a major complication of SARS-CoV-2 infection, stroke risk would be increased in the SARS-CoV-2 infected patients as well as post-infected patients with SARS-CoV-2. Although some limitations of data sources and others are existed, their results and conclusion are very acceptable. The information may contribute to understanding that the SARS-CoV-2 infection related hypercoagulable state is a major risk for the further serious complications including ischemic stroke.
Minor
In their conclusion in the last part of the paper, authors stated that “ This study presents a considerable number of young and majority male patients with SARS-CoV-2 and acute ischemic stroke who did not report vascular factors”. However, in my understanding, aged male patients may be a significant risk factor for SARS-CoV-2 and acute ischemic stroke”.
Please re-review your statement of conclusion.
Author Response
Reviewer Two:
Comments and Suggestions for Authors
It is presumably considered that the association between stroke and COVID-19 is probably multifactorial including an amalgamation of traditional vascular risk factors, proinflammatory and a prothrombotic state. Therefore, real-world collected data is required to reveal the mechanism and relationship between SARS-CoV-2 infection and stroke. In this point, the current study is very informative for many physicians and neurologists. The authors examined 323 acute ischemic strokes (AIS) patients from multicenter in 17 countries from the original dataset and 145 patients reported in the literature. Majority of patients suffered from an embolic-appearing stroke on imaging, and had about 50% risk of large vessel occlusions. The binominal logistic regression revealed that ischemic heart disease (OR, 4.9), atrial fibrillation (OR, 14.0), and active neoplasm (OR, 7.1) are associated with cardio embolic stroke. Since the hypercoagulable state is a major complication of SARS-CoV-2 infection, stroke risk would be increased in the SARS-CoV-2 infected patients as well as post-infected patients with SARS-CoV-2. Although some limitations of data sources and others are existed, their results and conclusion are very acceptable. The information may contribute to understanding that the SARS-CoV-2 infection related hypercoagulable state is a major risk for the further serious complications including ischemic stroke.
We appreciate the time and effort of reviewer two in reviewing our manuscript.
Minor
In their conclusion in the last part of the paper, authors stated that “ This study presents a considerable number of young and majority male patients with SARS-CoV-2 and acute ischemic stroke who did not report vascular factors”. However, in my understanding, aged male patients may be a significant risk factor for SARS-CoV-2 and acute ischemic stroke”. Please re-review your statement of conclusion.
We appreciate the comment. The association of aging and stroke, and also SARS-CoV-2 infection and older age have been well described in the literature. In a group of young patients without vascular risk factors and underlying comorbidities, SARS-CoV-2 might be the ignite of stroke. To address the concern of the reviewer, the authors revised the conclusion as bellow:
Page19, line125: Among patients with SARS-CoV-2 and acute ischemic stroke, there is a considerable number of young and majority male patients who did not report vascular risk factors. Therefore, young patients with SARS-CoV-2 infection should be monitors for the sign and symptoms of vascular events including ischemic stroke. It is reasonable to ensure that these patients and their families are aware of early signs of stroke (BE-FAST). (48) Stroke in other patients can be attributed to the existing comorbidity panel. We also observed that the proportions of comorbidities under each subclass of TOAST criteria were not different from the population studies prior to the SARS-CoV-2 pandemic.
Reviewer 3 Report
Dear Authors,
The study is very well done. The only comment I have - is that the potential clinical implication of these findings are not very clear to me. I would suggest to speculate a bit more on the subject, why this study is needed, and in which way the results can be useful for the clinical practice.
Author Response
Reviewer Three:
Comments and Suggestions for Authors
Dear Authors,
The study is very well done. The only comment I have - is that the potential clinical implication of these findings are not very clear to me. I would suggest to speculate a bit more on the subject, why this study is needed, and in which way the results can be useful for the clinical practice.
We appreciate the constructive comment by reviewer three. To address this issue, we modified different sections of the manuscript as follows:
Introduction:
Page 3, line 100-101: Understanding the population at risk for having a stroke after SARS-CoV-2 infection can promote timely diagnosis and proper management of these patients.
Discussion:
page 17, line 7-11: The results of our study indicated that SARS-CoV-2 infection can cause AIS among a considerable number of young and majority male patients who did not have vascular risk factors. The majority of these young patients had embolic-appearing stroke on their neuroimaging. Stroke in older patients can be attributed to the existing vascular risk factors.
Conclusion:
page 19, line 125-131: Among patients with SARS-CoV-2 and acute ischemic stroke, there is a considerable number of young and majority male patients who did not report vascular risk factors. Therefore, young patients with SARS-CoV-2 infection should be monitors for the sign and symptoms of vascular events including ischemic stroke. It is reasonable to ensure that these patients and their families are aware of early signs of stroke (BE-FAST). (48)
Reviewer 4 Report
Authors conducted multinational registry enrolling stroke patients following to COVID-19 infections, and systematic literature reviews. They clearly gathered stroke patients from their data base and systematic review. However, following issues are needed to be clarified.
Why dyslipidemia that was major risk factor of stroke was not involved in comorbidity?
Authors used specific methods such as EX-A, EX-S, ML-K, and ML-3. Although abundant data were analyzed by these methods, in Tables, there was little discussion about these. In particular, did authors intend that ‘a’ in EX-A or EX-S are stroke directly caused by COVID-19, while ‘b’ or ‘c’ were not? There was no discussion. Finally, difference between culprit and innocent bystander is unclear
GNSIS was suddenly appeared in the Discussion, which is very confusing together with the presence of p values. The contents in the methods, results, and discussion should match each other.
Line 80: cardiac patients can be revised by cardiovascular patients.
Line 96: it is necessary to demonstrate exact percentages of stroke caused by critical illness.
Author Response
Reviewer Four:
Comments and Suggestions for Authors
Authors conducted multinational registry enrolling stroke patients following to COVID-19 infections, and systematic literature reviews. They clearly gathered stroke patients from their data base and systematic review. However, following issues are needed to be clarified.
- Why dyslipidemia that was major risk factor of stroke was not involved in comorbidity?
We appreciate the comment. As reviewer four has mentioned, dyslipidemia is a major risk factor for stroke and cerebrovascular events. The multinational COVID-19 Stroke Study Group has collaborators from multiple centers in more than 30 countries across the globe. Data were collected based on a predefined protocol and validated through several steps. However, data regarding lipid profile and final diagnosis of dyslipidemia could not pass the quality control phase and authors decided not to include this comorbidity in analyses. We added this limitation to the manuscript as follows:
Page 19, line 113-114: In addition, we could not include dyslipidemia in the comorbidity list because data regarding lipid profile could not pass the quality control phase.
- Authors used specific methods such as EX-A, EX-S, ML-K, and ML-3. Although abundant data were analyzed by these methods, in Tables, there was little discussion about these. In particular, did authors intend that ‘a’ in EX-A or EX-S are stroke directly caused by COVID-19, while ‘b’ or ‘c’ were not? There was no discussion.
We appreciate the constructive comment by reviewer four. As the reviewer mentioned, we have included different models to answer the study aim. Overall, we deployed two main methods, clinical expert opinion (EX-A and EX-S) and unsupervised machine learning algorithms (ML-K and ML-S), to split the patients into different subgroups. In clinical subgroups, the patients are grouped based on the number of comorbidities. Thus, “a” means no comorbidity (EX-A3a, EX-S3a) or <2 comorbidities (EX-A2a, EX-S2a). In the ML models, we asked the machine to see the database and group the patients. We did not provide any assumption for this clustering (unsupervised). Thus, in ML-K and ML-S, a, b, and c are three distinct subgroups that machine could differentiate based on the similarity among these patients. In the next step, we aimed to see if clinical scoring models and machine learning algorithms could find similar subgroups. We realized that all models proposed three subgroups of patients who had stroke etiology and imaging patterns consistent with their comorbidity panel. As reviewer four has mentioned these comparisons are available in detail in the results section. The authors modified the discussion to better address the concern mentioned by reviewer four as bellow:
Page 17, line28:
The first group includes patients with no or very few comorbidities—EX-A3a, EX-S3a, ML-K3a, and ML-S3a. The majority of these patients are young men who had an embolic-appearing stroke. The second subgroup is distinguishable by having a high proportion of hypertension, diabetes, chronic kidney disease, and carotid stenosis, large-artery atherosclerosis origin of stroke, and embolic-appearing stroke on imaging—ML-K3b, ML-S3b, EX-A3b, and EX-S3b. The third group presented with hypertension, diabetes, ischemic heart disease, atrial fibrillation, congestive heart failure, smoking, and prior TIA or stroke—ML-K3c, ML-S3c, EX-A3c, and EX-S3c. The majority of the patients in the third group had cardioembolic strokes based on the TOAST classification and had a consistent imaging pattern. Subgroups of patients identified by clinical risk scoring and unsupervised clustering based on the comorbidity panels were similar in the original and literature review datasets. However, unlike the original dataset, the etiology of the stroke in the majority of patients in the second and third subgroups of the review datasets were reported as “strokes of undetermined etiology”. Overall, the identified pattern demonstrated by all models may indicate that AIS in only a subgroup of patients can be attributed to the SARS-CoV-2 infection (subgroup a in all models), while AIS in the second and third group of patients may be explained by the existing comorbidities
- Finally, difference between culprit and innocent bystander is unclear
We appreciate the comment. All patients included in this study had a stroke and all were positive for SARS-CoV-2 infection. In patients who have comorbidities and risk factors for stroke, stroke might be a consequence of the existing risk factors. But we cannot deny any possible role of SARS-CoV-2 infection in igniting stroke among these patients. Thus, the use of innocent bystanders may not be appropriate. However, if the patient has no known risk factor and is young, it is more probable that extra risk factors rather than the known comorbidity panel may cause the stroke (such as SARS-CoV-2 infection). In this manuscript, we proposed that SARS-CoV-2 might be considered as a culprit of stroke among these healthy and young patients. To address this concern by reviewer four, we have modified the discussion as follows:
Page 17, line 42-46: Overall, the identified pattern demonstrated by all models may indicate that AIS in only a subgroup of patients can be attributed to the SARS-CoV-2 infection (subgroup a in all models), while AIS in the second and third group of patients may be explained by the existing comorbidities
- GNSIS was suddenly appeared in the Discussion, which is very confusing together with the presence of p values. The contents in the methods, results, and discussion should match each other.
We appreciate the comment. GNSIS is the database of stroke patients in our home institute. Because there was no appropriate large-scale control group prior to the pandemic, the authors primarily decided to compare the finding of the current study with GNSIS. As reviewer four mentioned, the paragraph may seem confusing without the proper introduction of the database in the methods and result section. Accordingly, the authors decided to exclude this part from the discussion. The paragraph has modified as bellow:
Page18, line49; Our study indicated a subgroup of patients with no known comorbidities among the SARS-CoV-2 infected patients (22.0%). The result of our systematic literature review on SARS-CoV-2 infected stroke patients reported from 36 centers in 9 countries similarly demonstrated that 24% of the patients had no prior comorbidities.
- Line 80: cardiac patients can be revised by cardiovascular patients.
We appreciate the comment. The manuscript has been edited accordingly.
- Line 96: it is necessary to demonstrate exact percentages of stroke caused by critical illness.
We appreciate the comment. The authors modified the section as below:
Page3, line 97: As an example, there is an increased risk of ischemic stroke (OR>28) and hemorrhagic stroke (OR>12) within two weeks of sepsis. (26)This might be due to new-onset atrial fibrillation (6%) that puts the patient at risk of in-hospital stroke (2.6%).(24)
Round 2
Reviewer 4 Report
Thank you for revisions.
Authors appropriately revised their manuscript, and I have no more comments.